# Influence of Ablation Deformation on Aero-Optical Effects in Hypersonic Vehicles

**Bo Yang [1], He Yu [1,*], Chaofan Liu [1], Xiang Wei [1], Zichen Fan [2] and Jun Miao [3]**

1   School of Astronautics, Beihang University, Beijng 100191, China
2   Beijing Institute of Control and Electronic Technology, Beijing 100038, China
3   Qian Xuesen Laboratory of Space Technology, Beijing 100094, China
*   Correspondence: 13261059095@163.com

**Abstract:** High-speed turbulence is generated when hypersonic vehicles fly in the atmosphere, which can create aero-optical effects and interfere with optical navigation and guidance systems. At the same time, the front end and optical window of hypersonic vehicles are exposed to an aerodynamic heating environment, leading to the head ablation and thermal deformation of the optical window. This further aggravates the turbulent transition process and makes the error of the aero-optical effects more difficult to predict. In this paper, the aero-optical effects under the condition of high-temperature ablation were analyzed. Ablation deformation models of both the head and optical window were established. Then, a high-speed flow field was simulated under different flight conditions. The distortion characteristics of the aero-optical effects were obtained through the photon transmission theory. The simulation results show that the ablation deformation of hypersonic vehicles under an aerodynamic heating environment aggravates the disturbance error of the aero-optical effects. Moreover, with the increase in the flight speed and the decrease in the flight altitude, the ablation deformation of the hypersonic vehicles and the aero-optical effects distortion both gradually increase. The research in this paper provides a reference for the prediction of aero-optical distortion in an aerothermal environment.

**Keywords:** aero-optical effects; aerodynamic heating; ablation deformation; high-speed flow field; hypersonic vehicles





## 1. Introduction

In the context of the current demand for sophisticated weapons, research on hypersonic technology is experiencing a period of great development. Because hypersonic vehicles have a strategic significance in the attack–defense confrontation of future wars, many countries around the world are accelerating the research in this area [1,2]. Due to the high-accuracy requirements of hypersonic vehicles for information, optical navigation and guidance systems have become one of the most popular technologies [3]. However, when a hypersonic vehicle flies at high speed in the atmosphere, the high-speed flow field formed between its optical system and the incoming flow causes aero-optical effects, which will interfere with the optical detection system, causing the deviation, jitter, blur and energy loss of the received image [4,5]. Therefore, research on aero-optical effects is of great significance for the compensation of optical system errors in hypersonic vehicles.

The research on aero-optical effects mainly focuses on the acquisition of a high-speed flow field and the analysis of a disturbed light field. A high-speed flow field can be obtained via a wind tunnel test or a computational fluid dynamics (CFD) simulation [6,7]. A disturbed light field can be solved by the simulation of a ray tracing method of geometric optics or a micro-photon transmission method [8,9]. In recent years, many scholars have obtained the corresponding aero-optical distortion law through experiments and numerical simulations [10–12]. However, in addition to the irregular turbulence, severe heating is

generated between hypersonic vehicles and the air during high-speed flight, resulting in a huge thermal load on the surface of the hypersonic vehicles [13,14]. High-temperature air flow causes the head ablation and thermal deformation of the optical window [15,16], as shown in Figure 1. Therefore, in order to establish a more accurate law of the optical disturbance in an aerodynamic heating environment, it is necessary to study the distortion characteristics of the aero-optical effects while considering the ablation deformation.

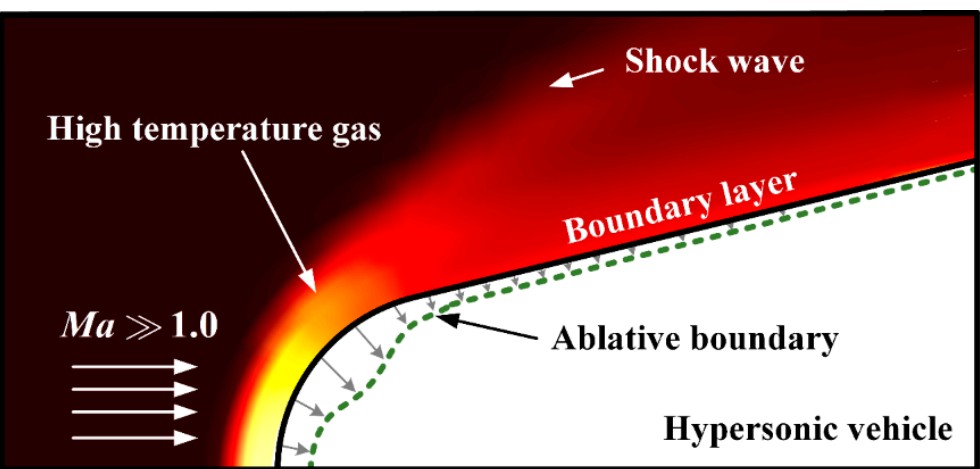

**Figure 1.** Schematic diagram of aerodynamic heating ablation of the hypersonic vehicle.

An aerodynamic heating environment not only causes the head ablation of hypersonic vehicles and the thermal deformation of the optical window but also leads to infrared thermal radiation interference. Because the main objective of this paper is to focus on optical distortion in the visible light band, the impact of infrared thermal radiation is not considered. The aerodynamic ablation process of hypersonic vehicles involves the coupled analysis process of aerodynamic heating, as well as the structural mechanism of heat conduction and ablation. Many scholars have studied the ablation profile using ablation experiments and numerical simulations [17,18]. A hypersonic vehicle entering the atmosphere generally has a certain ablation protective layer, and the change in its structural shape affects the distribution of the surrounding airflow [19]. In order to accurately capture the complex aerodynamic heating process that occurs during hypersonic flight, a moving grid algorithm with an implicit law of geometric conservation enhancement was proposed to measure the surface ablation and thermal response [20]. Some scholars have studied ablation characteristics and the impact of ablation on the hypersonic shock layer by combining experimental data with numerical simulations [21,22]. These high-temperature ablation studies of hypersonic vehicles can be used as the basis for obtaining the ablation profile of the head [23,24]. For the thermal deformation of the optical window, digital image technology is generally used to measure the displacement and strain at high temperatures [25]. Some scholars established an experimental system for measuring the deformation of the heated front surface of hypersonic vehicle components, aiming at the high-temperature flight environment. The measurement results for alumina ceramics show a good fit with the strain–temperature relationship based on Hillman's thermal expansion coefficient–temperature relationship [26]. A numerical simulation based on thermodynamics and structural mechanics is a powerful tool for the thermal deformation of optical windows [27]. It is the basis for obtaining the high-speed flow field in the subsequent aero-optical effects analysis.

In this paper, the large eddy simulation (LES) method is used to measure the high-speed flow field, as well as the transient structure of the hypersonic flow field [28]. In order to better analyze the optical disturbance law, this paper uses the previously proposed micromechanism of aero-optical effects to analyze the aero-optical distortion characteristics [9]. The key contribution of this paper is a thorough analysis of the aero-optical effects

of hypersonic vehicles in an aerodynamic heating environment with ablation deformation. An ablation deformation model of the head and optical window of a hypersonic vehicle is created to measure the ablation structure of hypersonic vehicles under different flight conditions (the head ablation and optical window deformation are mainly considered in this paper). The high-speed flow field under different flight conditions is obtained using the LES method. Finally, the distortion characteristics of the aero-optical effects are obtained via the photon transmission theory. The analysis of the aero-optical effects in this paper is similar to a real flight environment, which can help to develop aero-optical effects error compensation methods in an aerodynamic heating environment.

The remainder of this paper is organized as follows: In Section 2, the simulation of the head ablation and optical window thermal deformation is presented. The analysis and design of the aero-optical effects in an aerodynamic heating environment are described in Section 3. In Section 4, an aero-optical effects simulation of the hypersonic vehicles under ablation deformation is described and compared with the results for the ideal model conditions, and the distortion characteristics of the aero-optical effects under different flight conditions are analyzed. Finally, the conclusions are drawn in Section 5.

## 2. Simulation of Head Ablation and Optical Window Thermal Deformation

Large aerodynamic heat will be produced when hypersonic vehicles fly at high speed. High-temperature gas continuously transfers heat to the interior of a vehicle body during flight, resulting in surface ablation, especially at the head. This section describes the simulation of the ablation of the head and the thermal deformation of the optical window to provide a structural basis for subsequent high-speed flow field simulations.

### 2.1. Head Ablation Profile of Hypersonic Vehicles

As the ablation of a hypersonic vehicle head leads to a constant change in calculation coordinates, the instantaneous coordinate system $o_t x_t y_t z_t$ is generally used as an auxiliary solution. The origin $o_t$ of the instantaneous coordinate system is the vertex of the current front end, as shown in Figure 2. The initial state of the origin $o_t$ is the origin of the body coordinate system $o_b x_b y_b z_b$.

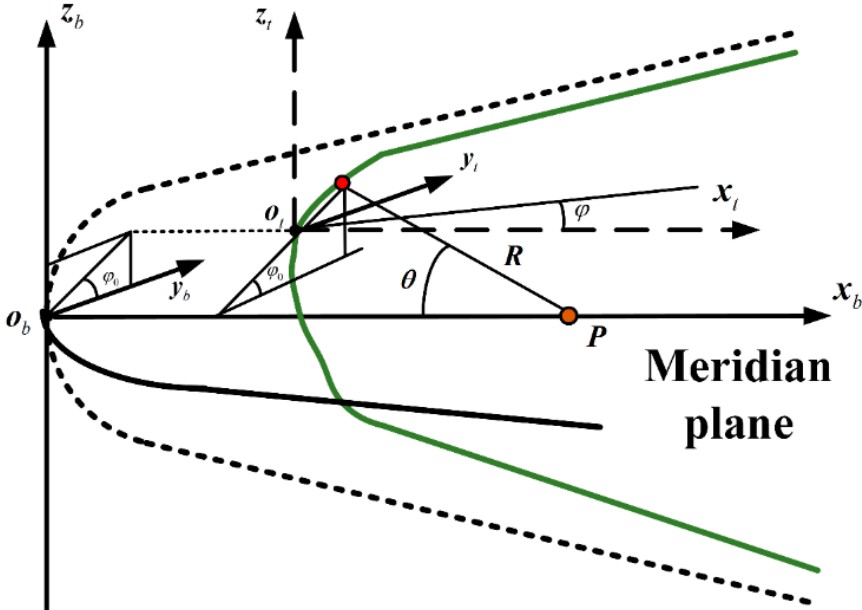

**Figure 2.** Schematic diagram of body coordinate system and instantaneous coordinate system.

With the occurrence of head ablation, the instantaneous coordinate system gradually changes. A point $P : (150\text{mm}, 0, 0)$ that cannot be ablated inside the vehicle is selected as the origin of the spherical coordinate system, and $R$ is the polar diameter from the origin $P$ to the ablation surface; $\theta$ and $\varphi$ are the spherical center angle and meridian angle, respectively. Therefore, the relationship between the instantaneous coordinate system and spherical coordinate system is as follows:

$$\begin{cases} x = x_P - R\cos\theta \\ y = R\sin\theta\cos\varphi \\ z = R\sin\theta\sin\varphi \end{cases} \tag{1}$$

The new $R$ of each meridian plane can be determined using an ablation calculation, and the coordinates $x, y, z$ of each point on the ablation surface can be calculated via Equation (1). Then, the point with the minimum $x$ coordinate on the entire ablation surface can be found by iterative updating. For the head of a hypersonic vehicle, the change rate of the ablation profile in the circumferential direction is far less than that in the longitudinal direction. Therefore, ablation deformation is usually treated as a two-dimensional problem, and the equation of the two-dimensional ablation profile is solved only for each meridian plane. The control equation describing the change process in the ablation profile in spherical coordinates is as follows [29]:

$$\frac{\partial R}{\partial t} = \dot{S}_P\left(\cos\theta + \frac{\sin\theta}{R}\cdot\frac{\partial R}{\partial\theta}\right) - V_{-\infty}\sqrt{1 + \left(\frac{1}{R}\cdot\frac{\partial R}{\partial\theta}\right)^2} \tag{2}$$

where $V_{-\infty}$ is the ablation rate of vehicle surface. Because point $P$ will not be ablated, the moving speed $\dot{S}_P$ of the coordinate origin satisfies $\dot{S}_P = 0$. When hypersonic vehicles fly in the atmosphere, the kinetic energy of the gas is converted into heat energy, and the heat flow enters the structure through the surface of the vehicle, causing a change in the temperature field inside the structure. According to the Fay Riddell equation, the heat flow at the stagnation point is as follows [29]:

$$q_{\omega,s} = 0.763 Pr^{-0.6}\left(\frac{\rho_\omega\mu_\omega}{\rho_s\mu_s}\right)^{0.1}\sqrt{\rho_s\mu_s\left(\frac{du_e}{dx}\right)}\cdot\left[1 + \left(Le^{0.52} - 1\right)\frac{h_D}{h_s}\right](h_s - h_\omega) \tag{3}$$

where $q_{\omega,s}$ is the heat flow of the stagnation point; $Pr$ is the Prandtl number, taken as 0.7; $\rho_\omega$ is the wall density; $\rho_s$ is the density of the stationary point; $\mu_s$ is the viscosity coefficient of the stationary point; $\mu_\omega$ is the wall viscosity coefficient; $u_e$ is the velocity immediately outside the boundary layer; $h_D$ is the dissociation enthalpy of air; $h_\omega$ is the wall enthalpy; and $Le$ is the Lewis number. $\rho_\omega$, $\rho_s$ and $u_e$ can be obtained through flow field calculation.

The heat flow in the non-stagnation point is treated by local similarity solution, and the approximate equation of laminar/turbulent heat flow on the surface of the aircraft head is obtained by using the method of flat-plate reference enthalpy:

$$q_{\omega,l} = 0.332 Pr^{-0.6}\rho^* u_e\frac{1}{\sqrt{Re_x^*}}(h_s - h_\omega) \tag{4}$$

$$q_{\omega,t} = 0.0296 Pr^{-0.6}\rho_e u_e Re_x^{-0.2}(h_s - h_\omega)\varepsilon \tag{5}$$

where $l$ and $t$ represent laminar and turbulent states, respectively. $\varepsilon$ is compressibility factor. $\rho^*$ and $Re_x^*$ are detailed in the literature [30].

The laminar/turbulent transition is analyzed by using the intermittent turbulence model, and the heat transfer calculation equation in the transition zone satisfies the following [29,31,32]:

$$q_{\omega,lt} = (1 - \Gamma)q_{\omega,l} + \Gamma q_{\omega,t} \tag{6}$$

where $l$ and $t$ represent laminar and turbulent states, respectively; $\Gamma$ is the intermittence factor, which has different expressions in different Reynolds number ranges [29,31].

$$\Gamma = \begin{cases} 0 & Re_\theta \leqslant Re_1 \\ \frac{1}{2}\left(\frac{Re_\theta - Re_1}{Re_2 - Re_1}\right)^2 & Re_1 < Re_\theta \leqslant Re_2 \\ 1 - \frac{1}{2}\left(\frac{Re_\theta - Re_3}{Re_3 - Re_2}\right)^2 & Re_2 < Re_\theta \leqslant Re_3 \\ 1 & Re_3 < Re_\theta \end{cases} \tag{7}$$

where $Re_1$ is the Reynolds number at the beginning of transition; $Re_2$ is the average Reynolds number; $Re_3$ is the Reynolds number at the end of transition; and $Re_\theta$ is the current Reynolds number. Therefore, it is assumed that $q_{\omega,j}$ is the heat flow of the aircraft surface, and the subscript $j = s, l, t, lt$ represents the heat flow density of different zones.

Actually, the ablation process is controlled by the gas boundary layer (external effects) and material properties (internal effects). The actual ablation process is very complex. Therefore, in engineering applications, the effective ablation enthalpy $H_{eff}$ is often used to couple these two effects, which is as follows:

$$\dot{m}_\omega = \frac{q_{\omega,j}}{H_{eff}} \tag{8}$$

where we assumed that $q_{\omega,j} = \alpha_0 \Delta I - \varepsilon_r \sigma_r T_\omega^4$ in order to simplify the operation. Item 1 is the sum of convective heating and recombined heating; item 2 is surface radiant heat. We assumed that $H_{eff} = 7500 + \eta \Delta I + (\xi - 1)h_{cw}$, where item 1 is the average sublimation enthalpy of carbon; item 2 is the enthalpy of thermal plug; item 3 is the enthalpy of mechanical denudation. More equation derivations are detailed in the literature [30].

The normal linear ablation rate of the hypersonic vehicle head is calculated from the effective enthalpy of the material. The head surface, considered as the heating surface, and the ablation rate equation for carbon/carbon composites is as follows [31]:

$$\begin{cases} \dot{m}_\omega = \frac{\alpha_0 \Delta I - \varepsilon_r \sigma_r T_\omega^4}{7500 + \eta \Delta I + (\xi - 1)h_{c\omega}} \\ V_{-\infty} = \frac{\dot{m}_\omega}{\rho_m} \end{cases} \tag{9}$$

where $\alpha_0$ is the convection heat transfer coefficient, taken as 500 W·m$^{-2}$·K$^{-1}$; $\Delta I$ is the term related to the wall enthalpy, which is the sum of convective heating and recombined heating; $\varepsilon_r$ is the emissivity, taken as 0.7; $\sigma_r$ is the Stefan–Boltzmann constant, taken as $5.6704 \times 10^{-8}$ W·m$^{-2}$·K$^{-1}$; $\eta$ is the thermal blocking factor, taken as 2.163; $\xi$ is the mechanical denudation factor, taken as 0.5; $T_w$ is the wall temperature, which is calculated by the equation of heat flux density and temperature gradient; $h_{cw}$ is the enthalpy of the cold wall; $\dot{m}_w$ is the mass loss rate per unit area of the material; and $\rho_m$ is the density of the material, taken as 3980 kg·m$^{-3}$.

This paper uses the hypersonic vehicle structure employed in previously published literature [9], as shown in Figure 3. $o_b x_b y_b z_b$ and $o_w x_w y_w z_w$ represent the vehicle body coordinate system and window coordinate system, respectively.

As the head ablation is the most serious in the actual aerodynamic heating environment, this paper mainly considers the front 100 mm of the vehicle for the ablation simulation. First, the heat flux density can be obtained by Equations (3)–(6) for the corresponding ballistic parameters, then the temperature distribution can be obtained by the relationship between the heat flux density and the temperature gradient in the heat transfer equation, then the ablation rate can be obtained by Equation (9) and then the ablation surface of the meridian plane can be calculated according to Equations (1) and (2). However, in the actual simulation process, in order to ensure the accuracy of the heat flow density, we use the commercial software Ansys Mechanical 2021R2 to conduct the heat conduction simulation. In order to facilitate the subsequent analysis of the aero-optical

effects on the head ablation, the ablation simulations were carried out at different flight altitudes and speeds, as shown in Table 1.

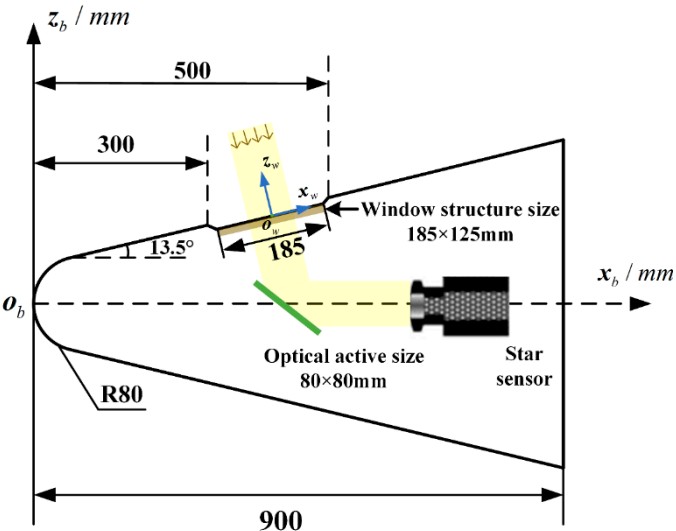

**Figure 3.** Physical structure diagram of a hypersonic vehicle [9].

**Table 1.** Different flight simulation conditions.

| Number | Altitude | Velocity |
| --- | --- | --- |
| 1 | 5 km | 3 Ma |
| 2 | 10 km | 3 Ma |
| 3 | 20 km | 3 Ma |
| 4 | 40 km | 3 Ma |
| 5 | 20 km | 2 Ma |
| 6 | 20 km | 5 Ma |
| 7 | 20 km | 10 Ma |

Because this study did not analyze heat ablation in detail, the attack angle was set as 0° without considering the changes in the attack angle, and flight time was 400 s during simulation. The longitudinal profile of the final head ablation was obtained as shown in Figure 4, which was the basis for the subsequent flow field calculation.

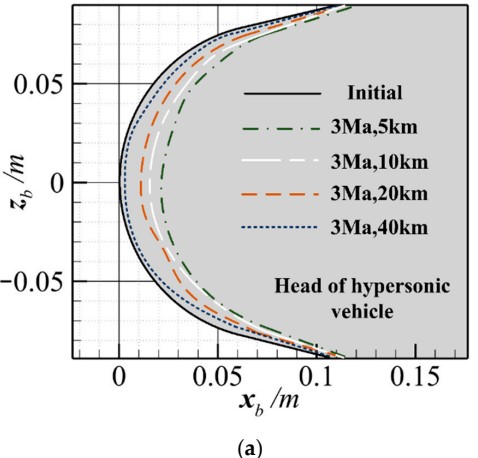

(**a**)

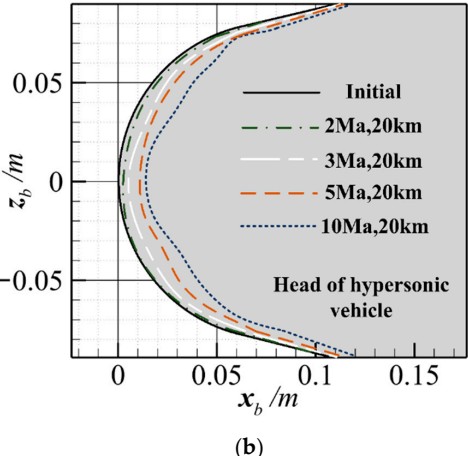

(**b**)

**Figure 4.** (**a**) Ablation profile at the same velocity and different altitudes; (**b**) ablation profile at the same altitude and different velocities.

*2.2. Thermal Deformation of Optical Window*

The thermal deformation of the optical window is also a coupling process of multiple physical fields, involving solid heat transfer, solid mechanics and material mechanics. In general, the calculation for the thermal deformation of the optical window includes two parts: solid heat transfer analysis and solid mechanics analysis.

During the flight time of the hypersonic vehicles, it is considered that the heat transfer process is stable and there is no internal heat source in the optical window. Therefore, the temperature distribution in the optical window satisfies the differential equation of steady-state thermal conduction without internal heat source, which is as follows:

$$\frac{\partial}{\partial x}\left(k_x \frac{\partial T}{\partial x}\right) + \frac{\partial}{\partial y}\left(k_y \frac{\partial T}{\partial y}\right) + \frac{\partial}{\partial z}\left(k_z \frac{\partial T}{\partial z}\right) = 0 \tag{10}$$

where $k_x, k_y$ and $k_z$ are the thermal conductivity values in the $x, y, z$ directions of the optical window, and $T$ is the temperature distribution in the optical window. To solve the equation, three kinds of boundary conditions, which correspond to each boundary of the optical window, are used. The first kind of boundary condition is the temperature distribution on a given boundary:

$$T = \widetilde{T}(\Gamma_1) \tag{11}$$

where $\widetilde{T}(\Gamma_1)$ is the temperature distribution of boundary $\Gamma_1$. The second kind of boundary condition is the heat flux on the specified boundary:

$$k_x \frac{\partial T}{\partial x} n_x + k_y \frac{\partial T}{\partial y} n_y + k_z \frac{\partial T}{\partial z} n_z = q(\Gamma_2) \tag{12}$$

where $n_x, n_y$ and $n_z$ are the direction cosine of the coordinate axis on the boundary pointing to the outside normal. $q(\Gamma_2)$ is the heat flux of boundary $\Gamma_2$. The third kind of boundary condition is the external ambient temperature and convection heat transfer coefficient on the given boundary:

$$k_x \frac{\partial T}{\partial x} n_x + k_y \frac{\partial T}{\partial y} n_y + k_z \frac{\partial T}{\partial z} n_z = h(T_a - T) \tag{13}$$

where $h$ is the convection heat transfer coefficient, and $T_a$ is the external ambient temperature of boundary $\Gamma_3$. When the heat transfer process of the hypersonic vehicles reaches a steady state, the temperature on the outer surface of the optical window is high and uneven, which makes the force on the window uneven and forms uneven stress and strain fields around the optical window.

According to the finite element calculation model of thermal deformation, and the boundary conditions given by the flow field simulation results, the thermal deformation of the optical window on hypersonic vehicles in an aerodynamic heating environment was simulated and analyzed using the multi-physical field simulation software Ansys Mechanical 2021R2. In this study, the material of the optical window on the hypersonic vehicle was sapphire crystal ($Al_2O_3$), and its physical properties are shown in Table 2.

**Table 2.** Physical properties of optical window [33].

| Physical Quantity | Value |
|---|---|
| Elastic modulus | 344 GPa |
| Poisson's ratio | 0.27 |
| Density | 3980 kg·m$^{-3}$ |
| Thermal conductivity | 36 W·m$^{-1}$·K$^{-1}$ |
| Constant pressure heat capacity | 750 J·kg$^{-1}$·K$^{-1}$ |
| Coefficient of thermal expansion | $5.3 \times 10^{-6}$ K$^{-1}$ |

According to the structure of the hypersonic vehicle in Figure 3, the optical window is modeled with a thickness of 5 mm. When the thermal deformation of the optical window is simulated by numerical methods, the optical window is discretized through a mesh subdivision, as shown in Figure 5. The grid elements and nodes of the optical window are 61,710 and 287,668, respectively.

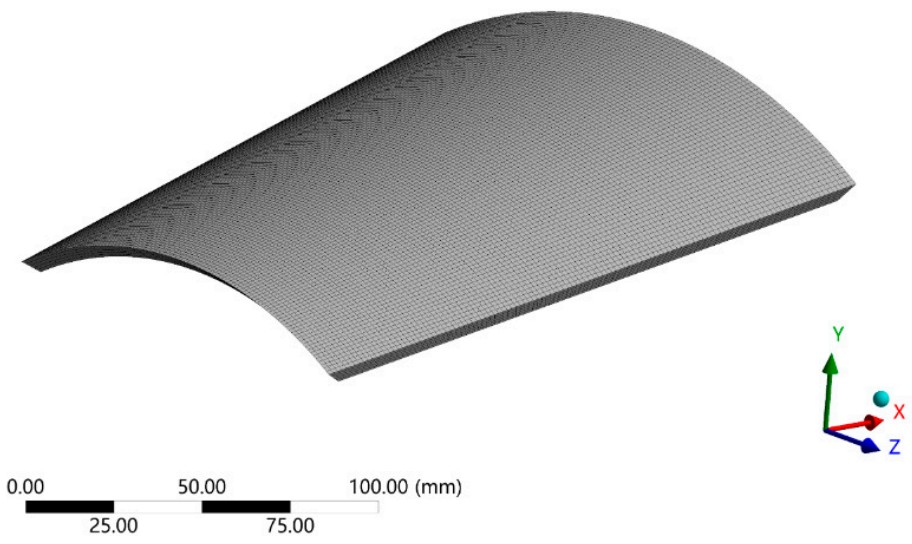

**Figure 5.** Mesh generation for thermal deformation simulation of optical window.

The boundary conditions need to be determined before the thermal deformation simulation of the optical window. The edges around the optical window are considered to be fixed to the hypersonic vehicle. Additionally, the boundary conditions of the upper and lower surfaces are set as shown in Table 3. The heat transfer coefficient $h$ is generally taken between 0 and 12 W·m$^{-2}$·K$^{-1}$ [34–36], which is taken as 10 W·m$^{-2}$·K$^{-1}$ in this paper. In addition, the temperature and the pressure were obtained through the steady-state conjugate heat transfer analysis and applied to the thermo-mechanical analysis to obtain the thermal deformation of the optical window.

**Table 3.** Boundary conditions of the thermal deformation.

| Boundary | Thermal Boundary | Solid Mechanical Boundary |
|---|---|---|
| Outer surface | The first kind of boundary (temperature distribution from external flow field) | Applied load (the pressure distribution from external flow field) |
| Inner surface | The third kind of boundary ($T$ = 300 K, $h$ = 10 W·m$^{-2}$·K$^{-1}$) | Applied load ($P$ = 101,325 Pa) |
| Window edge | The second kind of boundary ($q$ = 0) | Fixed constraint |

Under the given boundary conditions, different simulation conditions in Table 1 are used to obtain the thermal deformation results of the optical window, as shown in Figures 6 and 7.

It can be seen from Figures 6 and 7 that, with the increase in flight speed and the decrease in flight altitude, the thermal deformation of the window gradually increases, which affects the distribution of the high-speed airflow on the window.

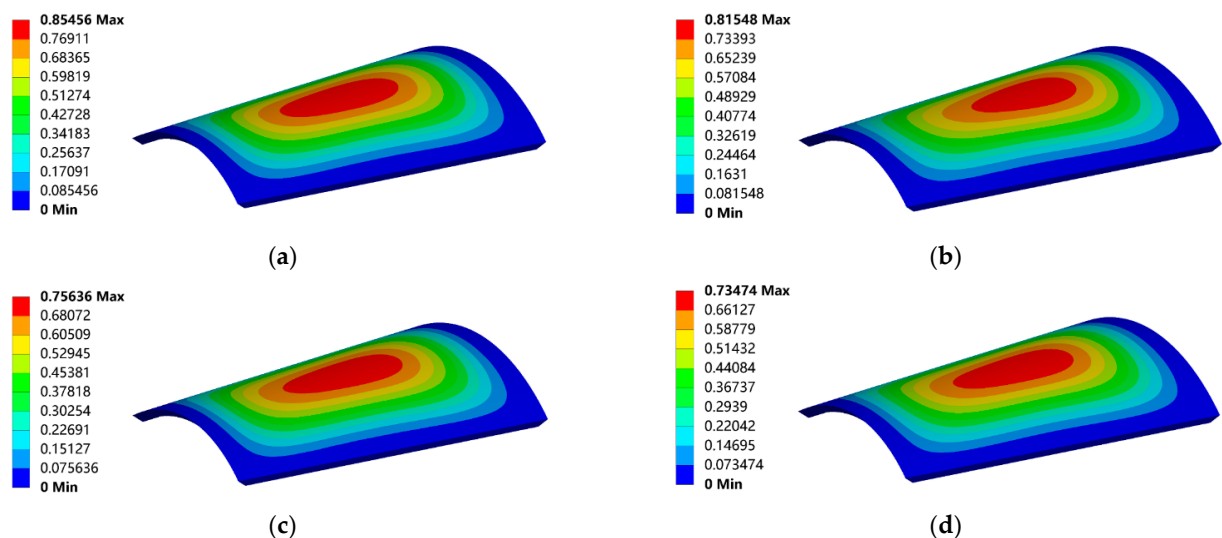

**Figure 6.** Thermal deformation (mm) of the optical window at the same speed and different altitudes: (**a**) 3 Ma, 5 km; (**b**) 3 Ma, 10 km; (**c**) 3 Ma, 20 km; (**d**) 3 Ma, 40 km.

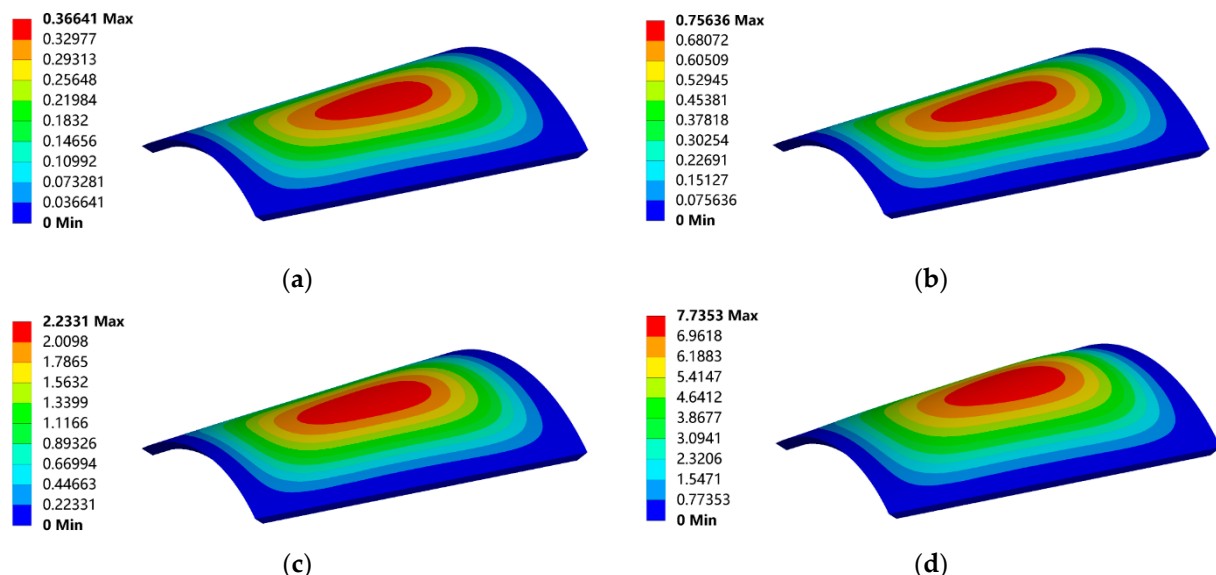

**Figure 7.** Thermal deformation (mm) of the optical window at the same altitude and different speeds: (**a**) 2 Ma, 20 km; (**b**) 3 Ma, 20 km; (**c**) 5 Ma, 20 km; (**d**) 10 Ma, 20 km.

## 3. Design of Aero-Optical Effects under Aerodynamic Heating Environment

This section describes a simulation of the aero-optical effects carried out in an aerodynamic heating environment. The thermal deformation structure (described in Section 2) was selected as the structure of the flow field simulation. The high-speed turbulence under different flight conditions was obtained through a large eddy simulation (LES), and then the distortion of the aero-optical effects is described based on the photon transmission theory.

### 3.1. High-Speed Turbulence of Structures with Thermal Deformation

In this study, the commercial software Ansys Fluent 2021R2 was used to simulate a high-speed flow field. Mesh generation is required before the flow field simulation. In order to more clearly reflect the impact of ablation, the grid was densified at the ablation boundary. Taking the simulation condition of 10 Ma and 20 km (as shown in Table 1) as an example, the grid distribution of the hypersonic vehicle is shown in Figure 8a.

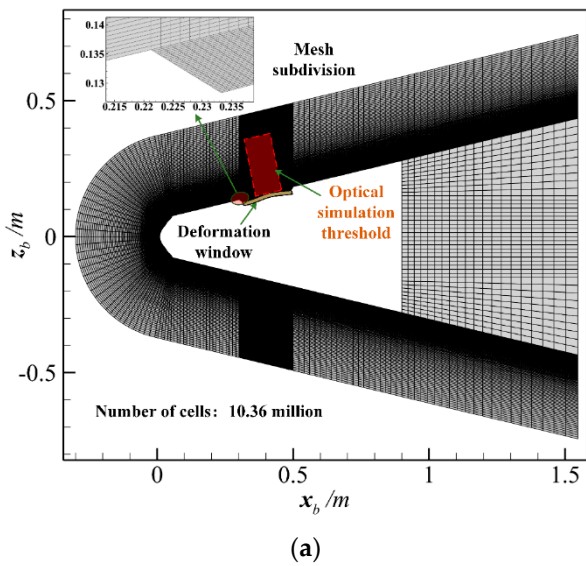

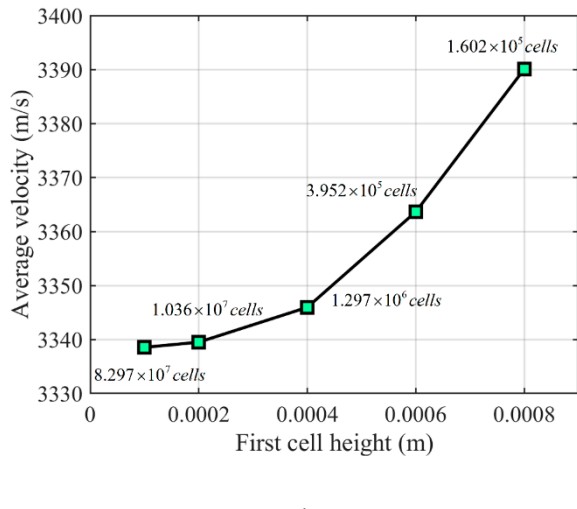

(**a**)                                                                                                          (**b**)

**Figure 8.** (**a**) Grid distribution of the hypersonic vehicle; (**b**) verification of the grid independence for the model.

The total number of the mesh generation is 10.36 million, and the thickness of the first layer of the optical window is 0.0002 mm. In order to verify the grid independence, a different number of grids is used for the LES simulation. The weighted average velocity of section $y_w - z_w$ above the window center is taken as the sampling standard. The very coarse grid ($1.602 \times 10^5 cells$), coarse grid ($3.952 \times 10^5 cells$), medium grid ($1.297 \times 10^6 cells$), fine grid ($1.036 \times 10^7 cells$), very fine grid ($8.297 \times 10^7 cells$) were employed to verify grid independence, as shown in Figure 8b, because the deviation of average velocity is less than 0.1% when using the fine grid. While ensuring accuracy and reducing the number of calculations, the number of grids is determined as 10.36 million. LES simulation was conducted for the simulation conditions (Table 1). The boundary conditions in the calculation were set as follows: The outlet was set as the constant pressure of the outlet condition; Because the thermal deformation results have been obtained in the previous steps, we did not use the dynamic grid but fixed wall condition in the LES simulation. However, in order to ensure the consistency with the previous temperature boundary conditions, the wall thickness is set to be 5 mm, the convective heat transfer condition is used for LES simulation and the calculation time step was taken as $10^{-7}$s. We used the WALE subgrid-scale model. The energy Prandtl number was 0.85, the wall Prandtl number was 0.85, the Cwale was 0.325 and the fluid was ideal gas. LES simulation steps are 40,000 and simulation time is 4 ms. In order to ensure calculation efficiency, a 256-core high-performance server was used for high-speed flow field calculations. Additionally, one of the obtained flow field results is shown in Figure 9a.

In the analysis process of gas flow, the change in vortex structure and density can reflect the fluid state, so we used two ways to analyze the turbulence state above the optical window. Firstly, the vortex structure is an intuitive reflection of the turbulent state. We use the conventional Q criterion to describe it, as shown in the Figure 9b. The change in the fluid velocity can be clearly seen through the vortex structure diagram of the Q criterion, and it also reflects the change in the turbulence state to a certain extent. It can be found that there is no vortex structure before the fluid enters the optical window, while the vortex structure gradually increases from the head to the tail of the optical window, representing the process of gradually developing from laminar flow to the turbulence.

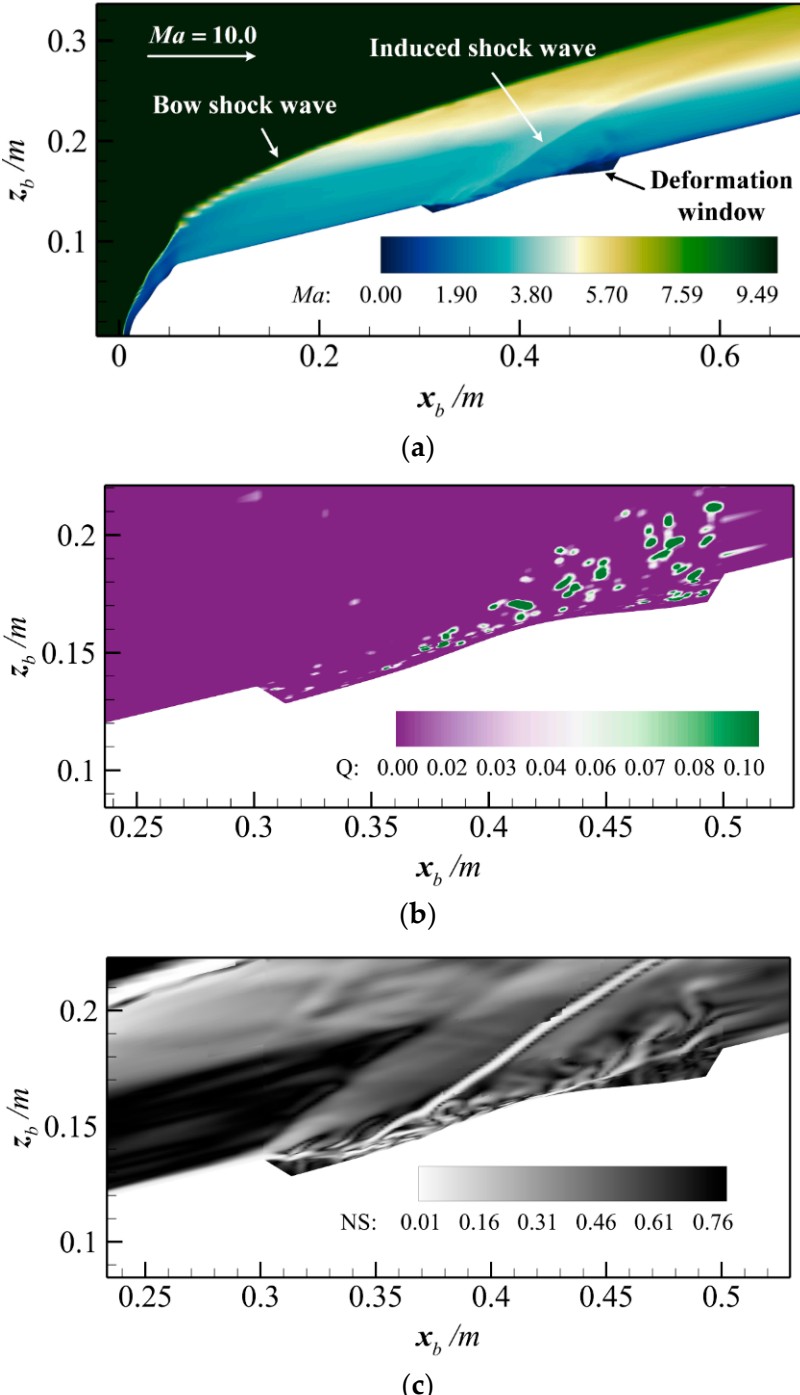

**Figure 9.** High-speed flow field under the condition of 10 Ma and 20 km: (**a**) Mach number of high-speed flow field; (**b**) the vortex structure of high-speed flow field; (**c**) the numerical schlieren of high-speed flow field.

In order to further explain the flow field state above the optical window, the schlieren method can be used to display the flow field details [37]. The expression of numerical schlieren is as follows:

$$NS = c_1 \exp[-c_2(k - k_{\min})/(k_{\max} - k_{\min})] \tag{14}$$

where $k$ is the density gradient amplitude, schlieren coefficient $c_1$ is 0.8 and $c_2$ is 60. The result of numerical schlieren is shown in Figure 9c.

It can be seen from the schlieren result that the shock wave is generated after the supersonic flow passes through the optical window, and the flow state in the boundary layer of the window is more disordered after the fluid is disturbed by the structure of the window. The change in the density gradient also shows that the fluid above the optical window is in a turbulent state. This paper is mainly concerned about whether the ablation structure will affect the calculation of the aero-optical effects. During the simulation process, the deformation process of high-temperature ablation is not considered, and only the final ablation structure is analyzed, so the chemical reaction of the air is ignored.

### 3.2. Description of Aero-Optical Effects Based on Photon Transmission Theory

After obtaining the high-speed flow field in the aerodynamic heating environment, an aero-optical effects analysis is required. The traditional method of analyzing a light disturbance mainly applies to the ray tracing method in geometric optics, which ignores scattering and absorption in the transmission process. In order to accurately reflect the actual aero-optical error, we used the MSAO method based on a micromechanism previously proposed to describe aero-optical distortion [9].

To facilitate the description of absorption, the refractive index $\widetilde{n}$ is expressed in the form of the Lorenz dispersion theory, which is as follows [38]:

$$\widetilde{n} = n_R + i n_I \tag{15}$$

where $n_R$ and $n_I$ are the real part and the imaginary part of the refractive index, respectively. According to the Gladstone–Dale law, we can calculate the relationship between the density $\rho$ and the real part of the refractive index, which is as follows:

$$n_R = 1 + K_{GD} \cdot \rho \tag{16}$$

where $K_{GD}$ is the Gladstone–Dale constant, and the relationship between $K_{GD}$ and wavelength $\lambda$ is as follows:

$$K_{GD} = 2.2244 \times 10^{-4} \left[ 1 + \left( 6.7132 \times 10^{-8} / \lambda \right)^2 \right] \mathrm{m}^3/\mathrm{kg} \tag{17}$$

The absorption and scattering process of photons was used in a previously proposed aero-optical microscopic mechanism [9]. In this paper, absorption and scattering coefficients are still used to describe the absorption and scattering of photons in turbulence, which are expressed as follows:

$$\mu_a = 2\nu n_I / c = \frac{N e^2}{4 m_e \varepsilon_0 c} \frac{\gamma}{(\nu_0 - \nu)^2 + (\gamma/2)^2} \tag{18}$$

$$\mu_s = N \sigma_s = \frac{8\pi}{3c^4} \left[ \frac{\pi^2 \left( n_R{}^2 - 1 \right)^2 \nu^4}{N} \right] \tag{19}$$

where the description of the relevant parameters can be found in the literature [9]. In this paper, $\overline{PDA}$ is used to describe the offset angle error caused by aero-optical effects, which is defined as the sum of the product of the weight of the photon number at any vector, $\boldsymbol{r}$, on the plane, $\Lambda$, and the deflection angle:

$$\overline{PDA}(\Lambda, \nu, t) = \sum_{\boldsymbol{r} \in \Lambda} PDA(\boldsymbol{r}, \nu, t) \frac{f(\boldsymbol{r}, \nu, \boldsymbol{\Omega}, t)}{\sum\limits_{\boldsymbol{r} \in \Lambda} f(\boldsymbol{r}, \nu, \boldsymbol{\Omega}, t)} \tag{20}$$

where $PDA$ is distribution of the photon offset angle on receiving plane; $\nu$ is the photon frequency; $f(\boldsymbol{r}, \nu, \boldsymbol{\Omega}, t)$ is the distribution function of the photons.

## 4. Results and Discussion

In this section, the simulation and analysis of the aero-optical effects in an aerodynamic heating environment are analyzed. The size of the optical sensor on the optical window was set to $80 \times 80$ mm$^2$, with an angle of view of $8°$. A rectangular flow field of $120 \times 120 \times 200$ mm$^3$ was selected as the optical simulation threshold in the window coordinate system, the same as in previous studies [9].

When using the MSAO method to simulate aero-optical effects, the photon computational domain must be divided. In this study, each flow field was divided into $N_x \times N_y \times N_z = 300 \times 300 \times 500$ equal parts based on a CFD grid, and the circumference angle was evenly divided into $360 \times 360$ angle units. The initial condition of the light source was set as a parallel light source. The initial incidence angle was $90°$; the wavelength was 572 nm; and the initial number of photons was $1 \times 10^8$. In order to explore the impact of the ablation deformation on the aero-optical effects under an aerodynamic heating environment, the aero-optical effects were simulated under the ablation deformation and the ideal model, respectively. Using the simulation condition of 10 Ma and 20 km in Table 1 as an example, two kinds of optical simulation thresholds are shown in Figure 10. There are 40,000 LES simulation steps, and the simulation time is 4 ms.

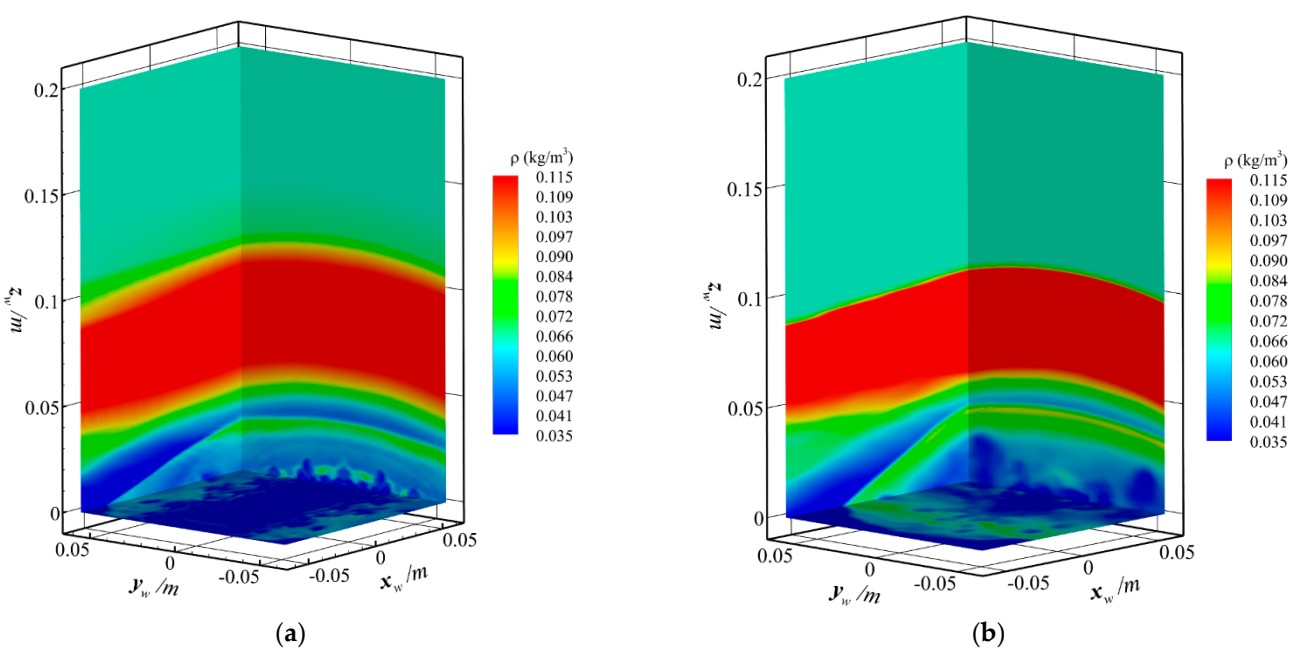

**Figure 10.** Comparison of photon simulation thresholds: (**a**) ideal model without deformation; (**b**) deformation model in thermal environment.

Figure 10 clearly shows that ablative deformation has a significant impact on the high-speed flow field structure on the optical window. The ablation of a hypersonic vehicle head causes the bow shock wave structure to be more compact and induces the shock wave to move backward. The gas density distribution is more concentrated above the optical window. In order to facilitate an exploration of the photon transmission process, different receiving planes were used to sample the photon offset angle $\overline{PDA}$. The receiving planes are from the bottom to the top of the photon simulation threshold and satisfy $z_w = 0 : 200$mm, $x_w \in (-40\text{mm}, 40\text{mm})$ and $y_w \in (-40\text{mm}, 40\text{mm})$ in the window coordinate system. The calculation results for the photon offset angle under the condition of different models are shown in Figure 11.

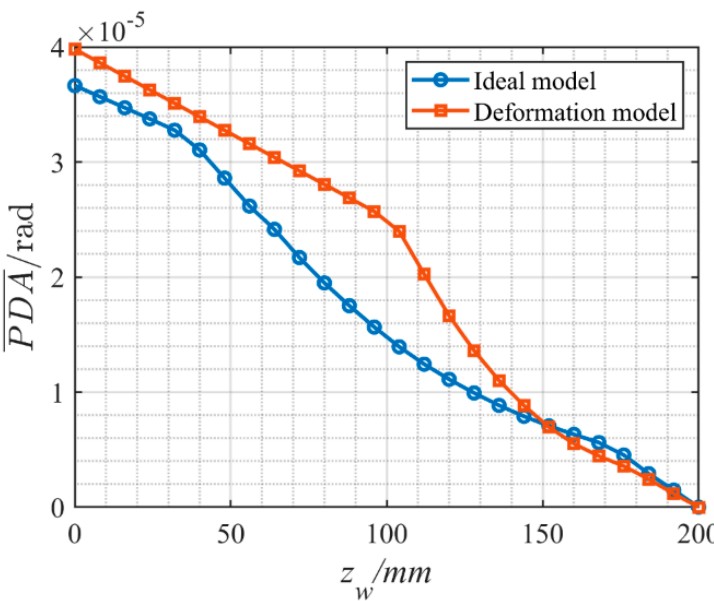

**Figure 11.** Comparison of two models for calculating the final offset angle.

Figure 11 shows that, with the increase in the photon transmission distance, the offset angle gradually increases, which conforms to the traditional rule of aero-optical effects. However, the offset photon angle under different receiving planes is clearly different, especially near the shock wave, which is consistent with the change in the actual flow field structure. Moreover, the aero-optical distortion caused by the ablation deformation model is larger. In order to conveniently describe the aero-optical error between the ideal model and the ablation deformation model, $\delta\overline{PDA}$ is used to describe the offset angle error, which is as follows:

$$\delta\overline{PDA} = \overline{PDA}_{deformation} - \overline{PDA}_{ideal} \tag{21}$$

where $\overline{PDA}_{ideal}$ and $\overline{PDA}_{deformation}$ are the simulation results of the aero-optical offset angle under the ideal model and ablation deformation model, respectively. Additionally, the simulation results under different flight conditions are shown in Figure 12.

Figure 12 also shows that the offset angle error and transmission distance caused by the different models actually satisfy the nonlinear relationship. The lower position in the curve is because it is in the vicinity of the shock wave, which will produce an inflection point. With the aggravation of the ablation in an aerodynamic heating environment, the final offset angle error also increases. The relative error is obtained according to the ratio of the error $\delta\overline{PDA}$ and $\overline{PDA}_{ideal}$. In addition, based on the average value of the relative error under seven different simulation conditions, it is estimated that the impact of the ablation deformation on the final aero-optical effects distortion is about 7.2%. To more comprehensively analyze the aero-optical effects in the thermal environment, simulations were conducted for different flight conditions, which are shown in Figures 13 and 14. There are 40,000 LES simulation steps, and the simulation time is 4 ms.

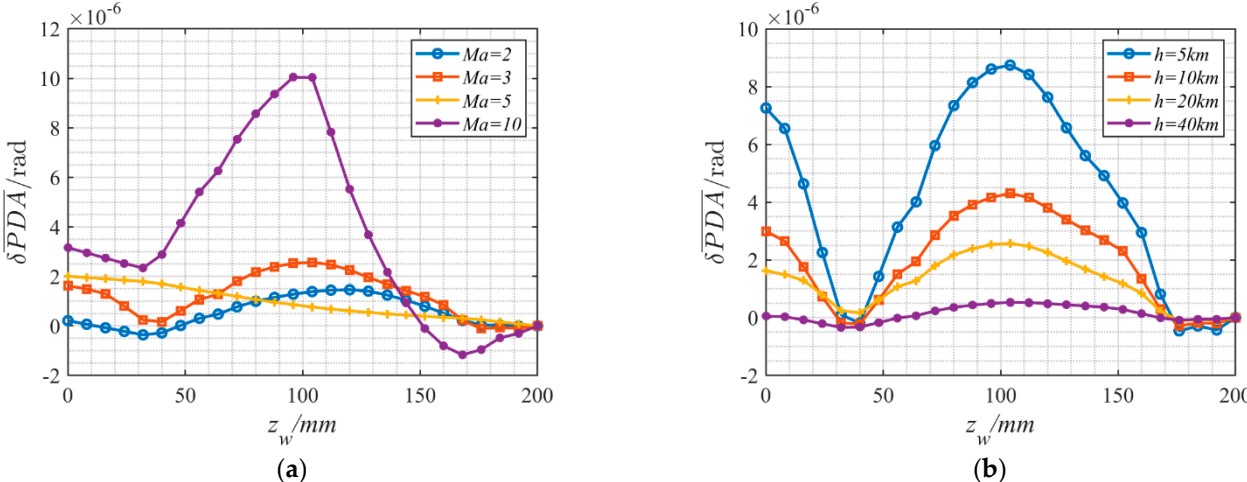

**Figure 12.** The offset angle error $\delta\overline{PDA}$ under ideal model and ablation deformation model: (**a**) the same altitude 20 km and different speeds (2 Ma, 3 Ma, 5 Ma and 10 Ma); (**b**) the same speed 3 Ma and different altitudes (5 km, 10 km, 20 km and 40 km).

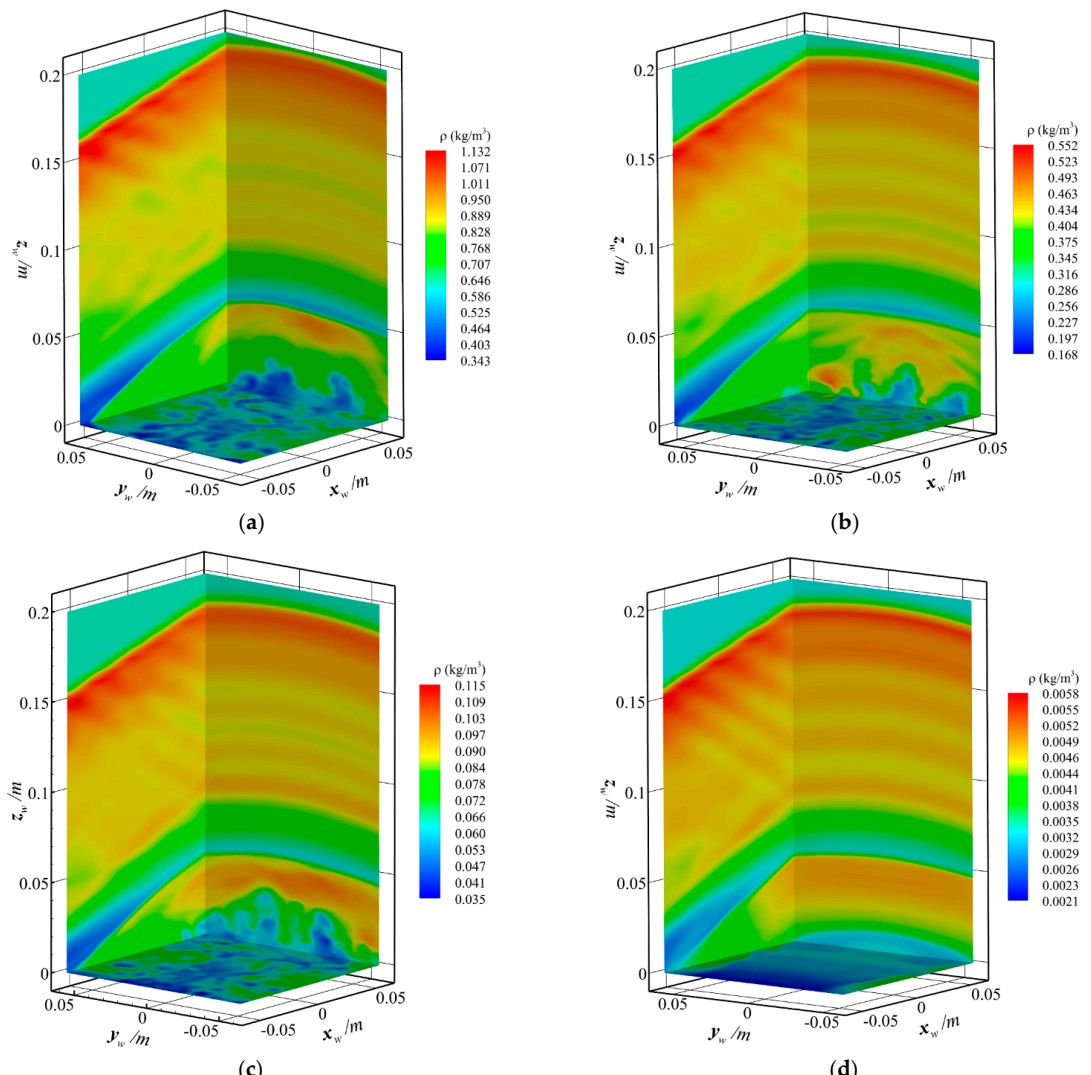

**Figure 13.** Optical simulation thresholds at the same speed and different altitudes: (**a**) 3 Ma, 5 km; (**b**) 3 Ma, 10 km; (**c**) 3 Ma, 20 km; (**d**) 3 Ma, 40 km.

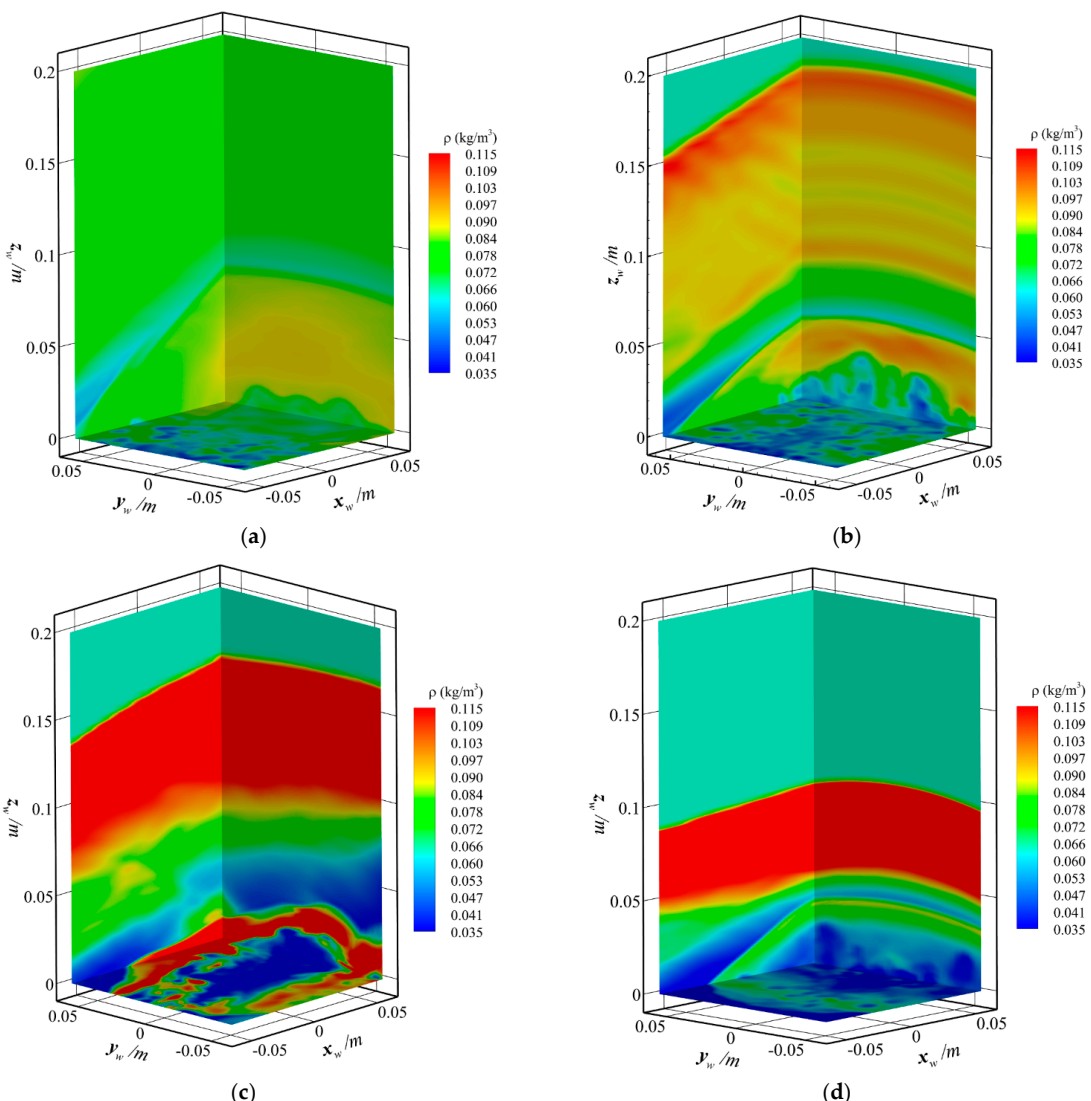

**Figure 14.** Optical simulation thresholds at the same altitude and different speeds: (**a**) 2 Ma, 20 km; (**b**) 3 Ma, 20 km; (**c**) 5 Ma, 20 km; (**d**) 10 Ma, 20 km.

Figures 13 and 14 show the optical simulation thresholds under different flight conditions. The change in the shock wave structure shown in Figure 13 is not as dramatic as that in Figure 14. Therefore, the velocity plays a decisive role in the angle of the shock wave. The higher the flight speed, the closer the shock structure is to the optical window, and the stronger the density compression at the shock, the greater the angle shift of the photon transmission.

From the perspective of the overall simulation threshold, the density change shown in Figure 13 is greater than that shown in Figure 14, indicating that the flight altitude plays a major role in the density distribution of the simulation threshold. This is mainly because the inflow density is different at different flight altitudes. With the increase in the altitude, the incoming flow density is lower. A photon transmission simulation is carried out for the flow field in Figures 13 and 14, and the results for the obtained aero-optical offset angle are shown in Figure 15.

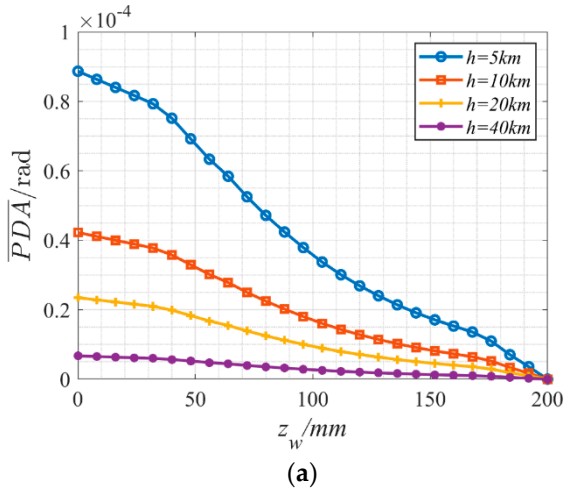 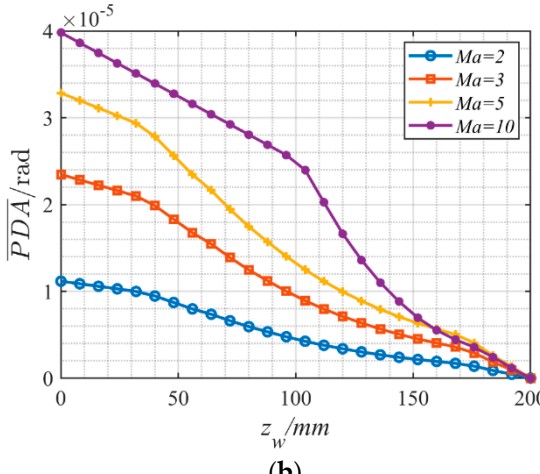

(**a**)　　　　　　　　　　　　　　　　　　　　　　　　　(**b**)

**Figure 15.** The offset angle $\overline{PDA}$ under different flight conditions: (**a**) the same speed and different altitudes; (**b**) the same altitude and different speeds.

It can be seen from Figure 15a that, at the same flight speed, as the flight altitude decreases, the offset angle of the aero-optical effects increases. This is because with the increase in the altitude, the air flow becomes increasingly thinner, causing a decrease in the perturbation effect of the high-speed flow field on the photon transmission. Similarly, Figure 15b shows that, at the same flight altitude, the offset angle of the aero-optical effects increases with the increase in the flight speed. This is because the increase in the flight speed and the effect of the aerodynamic thermal ablation aggravate the turbulence of the air flow above the optical window, which reduces the angle of the shock wave, resulting in greater air compression and an increase in the distortion of the aero-optical effects.

## 5. Conclusions

In this paper, the influence of the ablation deformation caused by the aerodynamic heating on the aero-optical effects was studied during the flight of hypersonic vehicles in the atmosphere. The simulation structure of a hypersonic vehicle is determined by establishing the head ablation model and the thermal deformation model of the optical window, and the aero-optical effects are simulated and analyzed through the high-speed flow field obtained by the LES method and the photon transmission theory based on a micromechanism. The simulation results show that the ablation deformation aggravates the distortion of the aero-optical effect, especially when the receiving plane is near the shock wave. Under the parameters of 10 Ma and 20 km, the aero-optical effects error of the ablation model is about 7.2% higher than that of the ideal model. Additionally, the ratio decreases following the decrease in the ablation deformation. Through an analysis of the aero-optical effects under different flight conditions, it can also be concluded that with the reduction in the flight altitude and the increase in the flight speed, the ablation deformation of hypersonic vehicles is enhanced, and the distortion error of the aero-optical effects increases.

**Author Contributions:** Conceptualization, B.Y.; methodology, H.Y. and Z.F.; project administration, H.Y.; validation, H.Y.; visualization, C.L.; writing—original draft, H.Y.; writing—review and editing, X.W. and J.M. All authors have read and agreed to the published version of the manuscript.

**Funding:** This research was funded by the Science and Technology on Space Intelligent Control Laboratory of China (No. ZDSYS-2018-03), the National Natural Science Foundation of China (No. 61973018) and the Civil Aerospace Technology Pre-Research Project of China (No. D040301).

**Institutional Review Board Statement:** Not applicable.

**Informed Consent Statement:** Not applicable.

**Data Availability Statement:** Data sharing is not applicable.

**Acknowledgments:** We thank the anonymous reviewers for their valuable comments which significantly improved the paper.

**Conflicts of Interest:** The authors declare no conflict of interest.

## Nomenclature

| | |
|---|---|
| $V_{-\infty}$: | Ablation rate of vehicle surface. |
| $\dot{S}_P$: | Moving speed of the coordinate origin. |
| $P$: | Stagnation point. |
| $R$: | Polar diameter from the origin $P$ to the ablation surface. |
| $\theta$: | Spherical center angle. |
| $\varphi$: | Meridian angle. |
| $q_{\omega,s}$: | Heat flow of the stagnation point. |
| $Pr$: | Prandtl number. |
| $\rho_\omega$: | Wall density. |
| $\rho_s$: | Density of the stationary point. |
| $\mu_s$: | Viscosity coefficient of the stationary point. |
| $\mu_\omega$: | Wall viscosity coefficient. |
| $u_e$: | Velocity immediately outside the boundary layer. |
| $h_D$: | Dissociation enthalpy of air. |
| $h_\omega$: | Wall enthalpy. |
| $Le$: | Lewis number. |
| $l$: | Laminar state. |
| $t$: | Turbulent state. |
| $\Gamma$: | Intermittence factor. |
| $Re_1$: | Reynolds number at the beginning of transition. |
| $Re_2$: | Average Reynolds number. |
| $Re_3$: | Reynolds number at the end of transition. |
| $Re_\theta$: | Current Reynolds number. |
| $\alpha_0$: | Convection heat transfer coefficient. |
| $\Delta I$: | Sum of convective heating and recombined heating. |
| $\varepsilon_r$: | Emissivity. |
| $\sigma_r$: | Stefan–Boltzmann constant |
| $\eta$: | Thermal blocking factor. |
| $\xi$: | Mechanical denudation factor. |
| $T_\omega$: | Wall temperature. |
| $h_{c\omega}$: | Enthalpy of the cold wall. |
| $\dot{m}_\omega$: | Mass loss rate per unit area of the material. |
| $\rho_m$: | Density of the material. |
| $k_x, k_y, k_z$: | Thermal conductivity values in the $x, y, z$ of the optical window. |
| $T$: | Temperature distribution in the optical window. |
| $\widetilde{T}(\Gamma_1)$: | Temperature distribution of boundary $\Gamma_1$. |
| $n_x, n_y, n_z$: | Direction cosine of the coordinate axis pointing to the outside normal. |
| $q(\Gamma_2)$: | Heat flux of boundary $\Gamma_2$. |
| $h$: | Convection heat transfer coefficient. |
| $T_a$: | External ambient temperature of boundary $\Gamma_3$. |

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
