# Peer review of "Influence of Ablation Deformation on Aero-Optical Effects in Hypersonic Vehicles"

_aerospace, doi:10.3390/aerospace10030232_

Round 1

Reviewer 1 Report

This study deals with aero-optical phenomena acting on hypersonic flight vehicles. The effects of changes in the shape of the head by aerodynamic heating and ablation on the aero-optical phenomena were analyzed with various flight conditions through computational simulations. As a follow-up study of the paper [9] that has already been published in this journal, Aerospace, the validation of the numerical techniques can be seen as almost completed. It is considered that the study method, computational results, and analysis for them are well described, and it is thought that it provides academic value that can be used in similar fields. However, the submitted manuscripts are rather ambiguous and have some descriptions that are difficult to be understood. If corrections and supplements are made to the following items, it can be recommended to be published in this journal.

- Please clearly describe the information for the simulation method or procedure. It seems that the computational simulations for the aero-optical analysis was performed for the final shape presented in Figure 4, is this correct? LES was performed on the final shape. A more detailed description of the numerical method, including the applied LES technique, is required. At what point in time are the results in Figures 10-15? These should be clearly and specifically explained.

- Is it reasonable to perform LES analysis? What is the unit Reynolds number of the inflow condition that simulates the flight condition? Since the simulations consider the practical applications, it is necessary to discuss whether the transition point from laminar to turbulent flow exists on the model.  Does the flow near the optical window correspond to turbulent flow? For the same reason, it seems necessary to consider the chemical reactions for the air, at least for the M10.

- Validations of the numerical methods for aero-optics and flow simulations can be confirmed from the published paper [9]. However, it is considered that another validations are required for the structural analysis of the shape change due to the ablation of the leading part or the thermal deformation of the optical window. Supplementation should be performed on the relevant matter. In particular, is h=10 W/m2-K of the thermal boundary of the inner surface applied in the structural analysis for the thermal deformation of optical windows reasonable? What is the basis? Apart from this issuu, is the optical window curved or flat? Curved surface is seen in Fig. 7, and in the results of Figure 10, etc., it looks flat. What is correct?

- Overall, the key to the presented results is that the greater the ablation by aerodynamic heating, the stronger the strength of the shock wave, and from this, the larger the aero-optical effect. The results are not at all new in the field of research related to aero-optics. In particular, the core of this study is to reasonably estimate the deformation of shape by aerodynamic heating, and to clearly connect the relationship with the aero-optical phenomenon that arises therefrom. The conclusion that the aero-optical effect is also greatly observed simply because the strength of the shock wave increases cannot provide great value.

-Minor corrections

- It is desirable to present a nomenclature. Also, the full name should be provided for the first occurrence of an abbreviation in the manuscript.

- A space is required before the unit notation.

- It would be better to explain more clearly what the results of Figures 10, 13 and 14 are. What is presented in the result is density. What does “Photon simulation threshold” mean? In particular, it is desirable to present the results in a different way so that all presented results can be seen. For example, the plane corresponding to x_w/m = 0.05 and y_w/m = 0.05 cannot be seen.

-page 13, line 2: It is questionable how the “linear relationship” described for Figure 12 results.

-page 13, line 6: How was the 7.2% derived? The explanation is very difficult to be understood.

Reviewer 2 Report

In this paper, the change of the aero-optic effect was investigated according to the thermal deformation of the optical window and the head ablation of hypersonic vehicles. 

Recently, hypersonic vehicles have attracted attention as next-generation airplanes and military weapons. Therefore, considerable efforts are given to developing hypersonic vehicles in several countries.

This paper can also be considered a part of these efforts. In this respect, potential readers of "Aerospace" might be very interested.

In this paper, a simplified method using recession velocity is applied to obtain the ablation profile of the head of a hypersonic vehicle.  

In addition, the deformation shape of the optical window was obtained through heat transfer analysis and structural analysis applying the given temperature and pressure boundary condition obtained by flow analysis.

The obtained head ablation profile and the optical window deformed shape were used in the LES simulation to investigate the aero-optical effect based on photon transmission theory.

The surface temperature of the vehicle, the ablation, the deformation of the vehicle, and the flow field are strongly connected and affect each other. Thus, an analysis considering the interaction between them is necessary to reflect the actual phenomena appropriately. 

However, the paper does not fully consider these interactions. Therefore, it is necessary to justify that the described method can reflect the actual phenomena to some extent in the revised manuscript.

More specifically, regarding the descriptions related to head ablation, the derivation procedure and connectivity between the equations are ambiguous. The explanation is fragmented, making it difficult to clearly understand the procedure for obtaining the head ablation profile. It is hard to know how the introduced physical quantities are utilized in the procedure.

Additionally, in the case of Equation 6, the adopted unit system and the numerical data are not presented adequately. So the information that can be obtained from the paper is incomplete and limited.

Regarding the thermal deformation of the optical window, it is stated that the temperature and pressure obtained from the flow field analysis are used as the boundary conditions. However, there is no clear explanation for this. It is not known whether this is the temperature obtained from the conjugate heat transfer analysis or the temperature and pressure obtained through some assumptions.

Since it is a very important issue related to the level of coupling considered in this paper, it should be dealt with in greater detail.

In addition, in the case of photon transmission theory, the previous paper of the authors was referred to, but it seems that the explanation is insufficient. Since the subject of this paper is to predict aero-optical effects, a more detailed explanation is needed, including figures.

Other minor comments are as follows.

For a clear understanding, it is recommended to make Fig. 2 in three dimensions.

In Figs. 10, 13, and 14, the range of the legend needs to be made the same for a clear comparison.

In Fig. 12, there are locations where the dPDA is reduced. The reason needs to be explained.

Reviewer 3 Report

The topic of this study is very relevant and can cause sufficient interest among researchers and designers of hypersonic aerospace vehicles, including hypersonic vehicles. Aero-optical distortion due to turbulence, strong shocks, surface ablation, non-equilibrium effects, ionization, etc. really can hamper communication and control of hypersonic flight. The paper is properly structured and contains the detailed description of numerical experiment. I think that the study can be recommended to publication in MDPI "Aerospace" journal. I am definitely not an expert in English, but I suppose that some lacks and typos (such as "ablation of head ablation, string 51) can be corrected in subsequent authors' work with editors.

Reviewer 4 Report

Review of the manuscript Aerospace-2106400, “Influence of Ablation Deformation on Aero-Optical Effects in Hypersonic Vehicles” by Bo Yang , He Yu, Chaofan Liu, Xiang Wei, Zichen Fan, and Jun Miao.
The article discusses a highly interesting topic, and its multidisciplinary nature makes it potentially useful research for the study of hypersonic flight. However, the various techniques used throughout the article are described in a summary manner and there is a complete lack of description on how they are harmonized in a sequence leading to the results. Therefore, while the research activity is very intriguing, the article does not allow other researchers to understand how the results were obtained. In a journal article, it is necessary for the authors to describe the methods used in such a way that their results are replicable by other researchers. This article, however, only accounts for the type of research conducted, provides some insight on the models adopted, displays the results, but omits the details. Many models are quickly described with some equations without even citing the sources from which they were drawn. It is more appropriate for a conference rather than a journal article. For this reason, I suggest that the article not be published unless the authors make substantial changes to the text, allowing the reader to understand how the techniques mentioned were utilized. In the following, the authors may find specific comments on certain parts of the text.
Line 49: I would say “”… severe heating “ rather than “… severe friction…”. The gas is mostly heated by compression through the shock wave rather than by wall friction.
Paragraph 2.1: how do you use the equations to obtain the final form of the object? In addition, which are the values that you used for all the parameters you mention?
Equation (2): provide a reference for this equation.
Line 132: strange use of semicolon
Line 139: what you call “stationary point” is normally defined as “stagnation point”
Line 140: “the velocity of the boundary layer” is “the velocity immediately outside the boundary layer”
Equations (3),(4), (5), and (6): provide references for these equations
Line 152: “Delta I is the term related to wall enthalpy”. Please specify better what it is.
Line 153: “Black degree” is called “emissivity”, and “the blackbody radiation constant” is the Stefan-Boltzmann constant
Lines 170-171: Could you better explain the procedure that brings to the calculation of the ablated profile? I assume you are using the equations you showed in this paragraph, but how, specifically? How do you obtain a local change of geometry?
Lines 231-232: some details about the CFD method? Is it a commercial code, a in-house code? Was it validated? Some references?
Lines 262-263: “…we use the photon transmission theory based on micro mechanism proposed previously to describe aero-optical distortion in this paper”. I missed where you previously proposed the method. Was it a reference, or else? If it is refence [9], please cite it here, too.
Figure 10a: The mesh seems to better capture the bow shock in the deformed case than in the undeformed case, where shock looks quite diffused. Difficult to say that "the bow shock moves downwards" if different grid resolutions are used.
Line 323. Specify which “same altitude and which “same speed”.

Author Response

Thank you very much for your kind review, it's very helpful, please see the authors' responses in the attachment.

Round 2

Reviewer 1 Report

It seems that the authors tried to revise and supplement the manuscript for most of the comments. However, the response to the point 2 remains unclear. The authors applied the LES model for turbulence and did not apply the boundary layer transition model. Nevertheless, the authors claimed that the simulation results indicate boundary layer transition has occurred. It is necessary to reconsider this. In addition, the answer to the point 2 must be clearly described in the manuscript.

Reviewer 2 Report

The revised paper looks better than the original paper. However, there are still some issues that should be clarified. The issues are as follows.

1) Regarding how to obtain the head profile due to ablation

1a) In the paper, the heat flow at the stagnation point was assumed through the Fay-Riddell equation. However, there is no explanation of how to obtain the heat flow at points other than the stagnation point. Detail explanation is required.

2b) In addition, the relationships between qws, q, ql, and qt are not appropriately described, making it difficult to know their roles.

3c) Further, there is no clear explanation of how the heat transfer analysis results are used to obtain the recession rate, including the DI. It should be explained in detail.

2) Regarding how to obtain the deformed shape of the optical window

In the paper, it is stated that a flow analysis was performed to obtain the thermal and pressure boundaries used for predicting the thermal deformation shape of the optical window.

However, it is not well presented how to obtain the thermal and pressure boundaries. It determines the level of how much the coupling effect between the surface temperature and the flow field is reflected.

Although it is not written in the paper, it seems that steady-state conjugate heat transfer analysis was carried out to obtain the temperature and pressure boundaries on the solid wall, when guessing through the author's answers and figures in the author’s reply (especially the "imported body temperature" indicated in the figure). The conjugate heat transfer analysis is a coupled analysis method that solves the heat transfer of solid and the flow field simultaneously.

2a) If the above guess is correct, it is strongly recommended to write on the paper that the temperature and the pressure were obtained through the steady-state conjugate heat transfer analysis and applied to the thermo-mechanical analysis to obtain the thermal deformation of the optical window.

2b) If not, the authors should explain in more detail how the temperature of the solid wall was obtained/or assumed.

3) Regarding the boundary condition in the LES simulation

3a) In the LES simulation, the adiabatic wall boundary condition was used, although a temperature boundary condition was enforced in the previous step calculation to obtain the thermal deformation of the optical window. Computation with the consistent boundary condition is required to guarantee the consistency of the analysis procedure.

Reviewer 4 Report

Review of the manuscript Aerospace-2106400, “Influence of Ablation Deformation on Aero-Optical Effects in Hypersonic Vehicles” by Bo Yang , He Yu, Chaofan Liu, Xiang Wei, Zichen Fan, and Jun Miao.

Round 2

The authors answered most points addressed by this reviewer in the first round of revision.

At this time, the paper might be published. I am just adding a couple of minor comments below:

Line 29: I guess the English language editor misunderstood the meaning of the sentence “A disturbed light field can be solved by rectified via a ray tracing method…”. I would change the sentence in “A disturbed light field can be solved by simulated via a ray tracing method…”.

In reference [1] the first author surname is "Di Giorgio", not just "Giorgio".

Round 3

Reviewer 2 Report

It seems that most of the raised issues were solved, and the revised paper looks more complete than the original version. This paper is recommended for publication in its present form.